# Live Yeast Supplementation in Gestating and Lactating Primiparous Sows Improves Immune Response in Dams and Their Progeny

**DOI:** 10.3390/ani12101315

**Published:** 2022-05-20

**Authors:** Tian Xia, Chenggang Yin, Marcello Comi, Alessandro Agazzi, Vera Perricone, Xilong Li, Xianren Jiang

**Affiliations:** 1Key Laboratory of Feed Biotechnology of the Ministry of Agriculture, Institute of Feed Research, Chinese Academy of Agricultural Sciences, Beijing 100081, China; summer7588@126.com (T.X.); ycg.ycg@foxmail.com (C.Y.); lixilong@caas.cn (X.L.); 2Department of Human Science and Quality of Life Promition, Università Telematica San Raffaele, Via di Val Cannuta 247, 00166 Rome, Italy; comimarcello@gmail.com; 3Department of Veterinary Medicine and Animal Science (DIVAS), University of Milan, Via dell’Università 6, 26900 Lodi, Italy; alessandro.agazzi@unimi.it (A.A.); vera.perricone@unimi.it (V.P.)

**Keywords:** live yeast, primiparous sows, weaning piglets, immunity

## Abstract

**Simple Summary:**

Primiparous sows are usually associated with poor lactation performance and low weaning weight piglets. Live yeast (LY) supplementation in gestation and lactation diets have been shown to improve sow health status, milk composition, and growth performance of suckling piglets. However, little is known about the carryover effects of LY supplementation during gestation and lactation on primiparous sows and their offspring. In the present study, LY supplementation significantly increased the serum concentrations of IgA and IgG of sows at farrowing and weaning stages, and of piglets at post-weaning on day 14 and 28. These results suggest that LY addition in the diets of gestating and lactating primiparous sows might improve the maternal and progeny health by increasing the immunity of sows and their offspring.

**Abstract:**

The present study determined the effects of live yeast (LY) supplementation during middle–late gestation and the lactation period in primiparous sows on reproductive parameters, lactation performance, and immunity, and also explores the carryover effects in their offspring. On day (d) 60 of gestation, 16 crossbred primiparous sows were randomly assigned to two dietary treatments (with or without supplementation of 425 mg/kg of live yeast; LYT and CT, respectively) homogeneous for body weight (BW) and backfat thickness. Experimental diets were applied from day 60 of gestation to the end of lactation. At weaning, 60 piglets with an average BW of each treatment were selected based on their source litter and assigned to two groups corresponding to the original treatments received by their mothers. Each group had five replicates of six piglets each and was fed a basal diet for 42 days. The results showed that LY supplementation significantly increased the serum IgA and IgG concentrations of sows at farrowing and weaning stages, and of piglets on day 14 and 28 post weaning. No significant differences were found in reproductive and lactation performance, while minor effects were observed on antioxidant capacity. In conclusion, live yeast addition during middle–late gestation and the whole lactation period resulted in enhanced immunity of primiparous sows and their offspring, therefore, improving maternal and progeny health.

## 1. Introduction

In the past several decades, sows have been successfully selected based on their reproductive performance, with an increasing number of piglets per litter. However, this led to the development of negative side effects, with hyperprolific sows, especially primiparous sows, bringing more and more weak newborn piglets into modern pig production. Additionally, primiparous sows are usually associated with poor maternal ability, such as poor lactation performance, high mortality of nursery piglets, and low piglet weight at weaning [1].

Measures had been taken to improve lactation performance of sows and growth performance of newborn piglets such as high nutritional diets, feeding and backfat thickness management, and other methods of modulating maternal and piglet health [2]. Probiotics have been considered as an effective antibiotic alternative to reduce pathogen infection and improve animal health, for which yeast is one of the most commonly used sources of probiotics [3]. Yeast supplementation has been known to improve the growth performance and immunological status of pigs [4,5]. Live yeast (LY) supplementation in gestation and lactation diets has been shown to improve sow health status, reproductive performance, and immune status of the progeny, although no or minor effects have been reported on milk composition, and growth performance [6,7,8].

Thus, the effect of LY supplementation in sow diet on their performance and health status is still inconsistent and needs to be further evaluated. Besides, little is known about effects of LY supplementation during gestation and lactation on growth performance of weaning pigs, especially in primiparous sows. Therefore, the present study was conducted to investigate the effects of supplementing LY in middle–late gestation and lactation diets on reproduction and lactation performance and immunological parameters of primiparous sows, in addition to the carryover effects on growth performance and immunity of the progeny.

## 2. Materials and Methods

The animal procedures in this study were approved by the Institute Animal Care and Use Committee of the Institute of Feed Research of Chinese Academy of Agricultural Sciences (FRI-CAAS-20210701). The trial was conducted at the Tianpeng experimental farm, located in Langfang, from July to November 2021.

### 2.1. Animals, Diets, and Management

On day (d) 60 of gestation, 16 healthy crossbred primiparous sows (Yorkshire × Landrace) that reached sexual maturity at 140~150 kg were inseminated in the second oestrus cycle and were randomly assigned to 2 homogeneous groups (CT, *n* = 8 and LYT, *n* = 8) based on their body weight (188.3 ± 6.4 kg) and backfat thickness (17.8 ± 0.22 mm). The experimental groups received the basal diet only (CT) or the basal diet supplemented with live yeast at 425 mg/kg (LYT, *Saccharomyces cerevisiae*, strain MUCL 39885, 1.5 × 10^11^ CFU/g). The basal diet (Table 1) was formulated to meet or exceed the nutrient requirements for primiparous sows according to National Research Council (NRC, 2012) [9]. The test material was provided by Prosol S.p.A. (Madone, Italy).

From day 60 to 107 of gestation, gestating sows were housed in the same room with individual pregnancy crates (2.2 m × 0.70 m) and received 2.5 kg (from d 60 to d 85 of gestation) or 3.0 kg (from d 85 to d 107 of gestation) of feed daily, partitioned in equal installments (07:00 and 15:00). On d 108, sows were moved to individual farrowing crates (2.2 m × 1.8 m). After farrowing, all sows were submitted to a step-up feeding regime, with the feed being supplied 4 times a day (07:00, 11:00, 15:00, and 18:00), starting at 2.0 kg/d and increased by 0.5 kg/d during the first 5 days; afterwards, all the sows were allowed *ad libitum* access to feed until weaning. When available, feed refusals were collected daily, and lactation feed intake subsequently calculated. During the first 24 h postpartum, cross-fostering within groups was performed, and litters were adjusted at 10 to 12 piglets per sow. All sows had *ad libitum* access to water for the whole experiment. The average temperature was maintained at around 22 °C in the gestation room, and about 25 °C in the farrowing room. Supplemental heat was provided to piglets with heat lamps (250 W). Two sows in the CT group and one sow in the LYT group were culled before parturition due to delivery and nursing problems.

After weaning, a total of 60 weaned piglets (d 28 ± 1, equally selected from each sows treatment, Duroc × Yorkshine × Landrace) with an average initial body weight of each treatment were selected based on their source litter and sex, and assigned to two groups corresponding to the original two treatments received by their mothers, with five replicates (pens) with 6 piglets per replicate. Both groups received the same basal pre-starter (Phase 1: d 0 to 14) or starter (Phase 2: d 14 to 42) diets. The basal diets were formulated to meet or exceed the nutrient requirements for weaned piglets (NRC, 2012) [9] for the respective rearing phases (Table 2). Piglets were housed in slatted floor pens (1.7 m × 1.5 m) and had *ad libitum* access to feed and water. The temperature of the nursery house was set at 28 °C during the first week and was then gradually adjusted to 26 °C.

### 2.2. Data and Samples Collection

Body weight and backfat thickness, at 65 mm to the right side of the dorsal midline at the last rib (P2) using an ultrasonic device (Renco Lean-Meatier; Renco Corporation, Minneapolis, MN, USA), were measured at d 60 of a sow’s gestation, at farrowing, and at weaning.

On d 1 and 28 of lactation, blood samples of primiparous sows were collected from the ear vein. Approximately 7 mL of blood per sows was collected into heparinized tubes (Greiner Bio-One GmbH, Kremsmünster, Austria), and centrifuged at 3000× *g* and 4 °C for 15 min (Eppendorf centrifuge 5810R, Hamburg, Germany). Serum samples were obtained from the supernatant and stored at −20 °C for later analysis.

Colostrum samples were collected manually from all the active mammary glands of one side within 1 h after the onset of farrowing, while milk samples were collected on d 14 of lactation, after an intravenous injection of 2.0 mL oxytocin (Hangzhou Animal Medicine Factory, Hangzhou, China) via the ear vein. The samples were immediately stored at −20 °C until the further analysis.

At parturition, total born, born alive, still born, and mummies piglets were recorded for each sow. Individual body weight of piglets was determined within 24 hrs from farrowing, at weaning, and on d 14, 28, and 42 post-weaning. Litter weight and average daily gain were subsequently calculated accordingly. Post-weaning replicate feed intake was determined for periods 0–14, 14–28, and 28–42, and relative feed efficiency calculated.

The diarrhea incidence of each piglet was recorded at the same time every morning during the first two weeks of the post-weaning trial, and the incidence of diarrhea (%) was expressed as the percentage of piglets with diarrhea related to the total number of weaned piglets according to the method of Zhao et al. (2021) [10].

On d 0, 14, and 42 post weaning, blood samples (about 7 mL) were collected from the same piglet in each pen via the jugular vein. The selection of the piglets occurred at d 0, selecting within each pen the animal whose body weight was closer to the pen average. The blood samples were centrifuged at 3000× *g* and 4 °C for 15 min to get the serum, which was stored at −20 °C until analysis.

### 2.3. Determination of Milk Compositions

Colostrum and milk samples were analyzed for the concentrations of protein, fat, lactose, and solids-not-fat by means of an automatic milk analyzer (Milk-Yway-CP2, Beijing, China). The results are expressed as the percentages in colostrum and milk.

### 2.4. Determination of Immunological and Antioxidant Parameters in Serum

Concentrations of immunoglobulin A (IgA), immunoglobulin G (IgG), and immunoglobulin M (IgM) in the serum of sows and piglets were detected by corresponding commercial kits (Bethyl Laboratories, Inc., Montgomery, TX, USA). Methods were used according to the description of Zhao et al. (2017) [11] with slight modifications. Briefly, flatbottomed 96-well micro-titer plates were coated with a 100 μL ovalbumin solution (25 μg/mL) and incubated overnight at 4 °C. After three washes, the wells were blocked with 5% skimmed milk and incubated for 1 h at 37 °C. After a further three washes, 100 μL of serum (diluted 1:80) was added to each well. The plates were then incubated for 1 h at 37 °C, followed by three more washes; 100 μL of horseradish-peroxidase conjugated rabbit anti-swine immunoglobulins was added to each plate. The plates were further incubated for 1 h at 37 °C and washed three times. The substrate solution (150 μL) was added to each well. The plate was incubated for 15 min at 37 °C, and the reaction was terminated by 50 μL H_2_SO_4_ (2 mol/L). The optical density was read at 450 nm. Standard curves were generated using serial dilutions of the recombinant porcine immunoglobulin. The results were expressed in mg/mL based on a standard curve. Levels of the malondiadehyde (MDA) and total antioxidant capacity (T-AOC), and activities of superoxide dismutase (SOD) and catalase (CAT) in serum were measured by a spectrophotometer (Lengguang SFZ1606017568, Shanghai, China), according to the instructions of the corresponding reagent kits (Nanjing Jiancheng Institute of Bioengineering, Nanjing, China). Briefly, MDA concentrations were determined using 2-thiobarbituric acid and the optical density (OD) value was read at 532 nm. T-AOC levels were determined by the ABTS method, and the OD value was collected at 405 nm. SOD activities were calculated through a nonenzymatic nitroblue tetrazolium (NBT) test, which measures the inhibition of the formation of superoxide anion free radicals that reduce the nitroblue tetrazolium of the sample, and the OD value was read at 450 nm. CAT activities were measured with ammonium molybdate, and the change in absorbance was recorded at 405 nm.

### 2.5. Statistical Analysis

Data were analyzed using the general linear model (GLM) procedure of SAS v. 9.2 (SAS Inst. Inc., Cary, NC, USA). Primiparous sows, their litters and pens (replicates) were served as the experimental units in the model, which included diet (CT or LYT) as the main effect. For the analysis of reproductive performance, sows and their litters were used as the experimental units. For the analysis of milk composition, immunological and antioxidant parameters in the serum of primiparous sows and piglets, sow or piglet was used as the experimental unit. Treatment means were calculated using the LSMEANS statement and separated by the Student–Newman–Keuls test. Moreover, the chi-square test was used to analyze diarrhea incidence. Differences were declared significant at *p* ≤ 0.05, while 0.05 < *p* ≤ 0.10 was considered to indicate a trend in the data.

## 3. Results

### 3.1. Body Weight, Backfat Thickness, and Reproductive Performance of Primiparous Sows

Results showed that a LY diet during middle–late gestation and lactation period did not significantly alter body weight, backfat thickness, and average feed intake of sows (Table 3). Similarly, reproductive performance of primiparous sows was not affected by LY supplementation (Table 4), as evidenced by the total number of piglets born, still born, mummy, and individual birth weight or litter birth weight, which were all similar to the CT group (*p* > 0.05). However, born alive and percentage of average birth weight < 1.0 kg in the LYT treatment were numerically lower than those in the CT group (10.57 vs. 13.00, and 2.22 vs. 10.55, respectively), but showed no significant difference (*p* > 0.05). Additionally, the number of weaned piglets, litter weight or individual weight at weaning, and average daily gain of sulking piglets during lactation were not significantly influenced by dietary treatments (*p* > 0.05). The LY diet numerically improved average daily gain of sulking piglets by 11.89% with 228.8 g/day during the whole lactation period (*p* > 0.05).

### 3.2. Growth Performance and Diarrhea Incidence of Weaned Piglets

The carryover effects of dietary supplementation with LY during middle–late gestation and the lactation period in primiparous sows on weaning piglets are reported in Table 5. Although average body weight (BW) of piglets at the end of lactation was not significantly different between CT and LYT groups (7.25 vs. 7.96, *p* > 0.05, Table 4), the selection of the piglets for the post-weaning trial resulted in a significant different BW at day zero, with LYT piglets being heavier than CT (*p* = 0.02). Average daily gain (ADG), average daily feed intake (ADFI), and gain: feed ratio (G: F) during the post-weaning were not significantly affected on the basis of the dietary treatments received by the mothers (*p* > 0.05), although ADFI in LYT tended to be 32.12% higher than those in the CT group (719 vs. 950, *p* = 0.10) from day 28 to 42. The percentage of diarrhea incidence during post weaning day 0–14 in the LYT tended to be lower than that in the CT group by 29.93% (16.93 vs. 13.03, *p* = 0.10).

### 3.3. Colostrum and Milk Compositions

As shown in Table 6, there were no significant differences in the percentages of fat, protein, lactose, total solids and solids-not-fat in the colostrum, and milk between primiparous CT and LYT sows (*p* > 0.05). Only milk lactose tended to be higher in the LYT group compared to the CT group (*p* = 0.10).

### 3.4. Immunological Responses

Effects of LY supplementation during middle–late pregnancy and the lactation period on serum immunological parameters in primiparous sows and weaned piglets are presented in Table 7. Serum concentrations of IgA in sows at farrowing and at weaning, and in piglets on day 0, 14, and 42 post-weaning in the LYT group were significantly higher than those in the CT group (*p* < 0.05). Similarly, serum IgG concentrations in sows at farrowing and at weaning, and in piglets on day 14 and 42 post-weaning were increased in the LYT group compared to the CT group (*p* < 0.05). Additionally, serum IgM concentration in LYT piglets was higher compared to the CT group at weaning (*p* < 0.05) and showed the same trend at day 14 post-weaning (*p* = 0.06).

### 3.5. Antioxidant Responses

Effects of LY supplementation during middle–late pregnancy and the lactation period on serum antioxidant parameters in primiparous sows and weaned piglets are shown in Table 8. Compared with the CT treatment, serum activity of CAT in sows at weaning was improved by LY supplementation (*p* < 0.01). However, no significant differences were found in serum concentrations of MDA and T-AOC, and serum SOD activity in primiparous sows and in weaned piglets between CT and LYT (*p* > 0.05). Only SOD tended to be higher in LYT piglets at day 14 post-weaning (*p* = 0.10).

## 4. Discussion

The present study aimed to evaluate the effect of LY administration during middle–late gestation and lactation on primiparous sows and their progeny. In our trial, reproductive performance was not affected by LY supplementation. Body weight and backfat thickness of primiparous sows at day 110 of gestation and 28 days postpartum were not affected by dietary treatment. Similar results have been observed in other studies, where the inclusion of LY in sows diets, irrespectively from the administration period (i.e., from day 90 of gestation to weaning, or the whole gestation and lactation period), did not result in any significant difference in body weight [6,8] or backfat thickness of multiparous sows at 110 days of gestation and 21 days postpartum [8,12]. These results are consistent with the lack of significant treatment effect on feed intake [8,12], which was also observed in our study. However, Domingos et al. (2021) [13] found that LY administration in the lactation diet of sows improved daily feed intake and total feed intake. The discrepancies in the observed results may be attributable to different factors, such as the number of animals included in the trial, parities of sows, LY strains, and seasons. Total number of piglets born, born alive, still born, and mummies were not affected by dietary treatment in this study, which is consistent with the results of previous research [6,8,12,13,14]. It was reported that the litter size in swine was mainly determined by the fertilization rate and prenatal death occurring in the early pregnancy [15], and was not likely to be influenced by diets during late gestation, thus suggesting that live yeast administration in the middle–late gestation had no effect on the survival rate of embryos and fetuses.

In line with the reproductive performance, lactation performance was also not affected by LY administration. No significant differences were found in the percentages of fat, protein, lactose, and total solids and solids-not-fat in the colostrum and milk on lactation on day 14 between CT and LYT sows. Similar results were reported in the literature, where colostrum and/or milk composition were not affected by LY administration [8,12]. However, this result contrasts with the works of Jurgens et al. (1997) [6], who demonstrated a significant increase of total solids and crude protein in milk of sows, and of Peng et al. (2020) [12], who found the concentrations of protein, lactose, and solids-not-fat in colostrum were significantly higher in LY-supplemented sows. The underlying mechanism may be associated with the optimization of bacterial profile in the gut, which would enhance the utilization of nutrients for colostrum or milk composition [16]. In fact, the amount of colostrum and milk intake may be more important than the composition for the growth of piglets during the suckling period [17].

Like sows’ reproductive and lactation performance, growth performance of piglets during lactation and in the post-weaning phase were also not influenced by LY supplementation. Individual weight and litter weight at birth and at weaning, and ADG of suckling piglets during lactation were not significantly different between piglets born from LTY and CT sows. This result is in agreement with previous studies, which showed that LY supplementation in the diet of sows did not affect the weight of the litter and piglets at birth and weaning, as well as mean weight gain of litter and piglets during lactation [6,8,12]. However, other studies reported increased average weaned piglets per litter and weaning litter weight were higher in sows receiving LY [18]. In a study conducted by Domingos et al. 2021 [13], although the treatments did not influence the average birth weight and average litter weight at birth, the diets containing LY showed higher piglet weight at 14 day. Additionally, the same authors observed that the use of LY improved average daily weight gain of the piglets and litter daily weight gain [13]. Peng et al. (2020) [12] reported that sows receiving LY in the diet had a lower number of low body weight piglets compared to sows fed the CT diet. This result is in line with our findings, where the percentage of average birth weight < 1.0 kg or < 1.1 kg in the LY treatment were numerically lower than those in the CT treatment.

In the post-weaning phase, ADG, ADFI, gain: feed of piglets were not significantly affected by the dietary treatment received by their mothers, but ADFI during post weaning in days 28–42 in LYT tended to be higher than those in the CT treatment by 27.48%. Similarly, although not significantly different, LYT ADG was numerically higher than CT by 32.12%. This is consistent to the report by Lu et al. (2019) [14], where the ADG from weaning to days 21 and 42 post-weaning was increased by LY supplementation in the gestation and lactation diets; however, in that study, no effect on post-weaning feed intake was observed. Furthermore, supplementation with LY during gestation and/or lactation period enhanced piglet performance and feed intake [4,19,20,21,22]. The addition of LY and yeast extracts had no other effects on any further growth performance or feed intake in weaned piglets [23,24].

Although in the present study there are no supporting data, it is well known that one the mechanism of action of LY is the modulation of the gut microbiota [25]. It is possible to speculate that in the present trial, LY administration during gestation and lactation modulated the maternal gut microbiota, which was later transferred to the offspring. Such a hypothesis is supported by our finding that diarrhea incidence during post-weaning on days 0–14 in the piglets born from dams receiving LY was reduced compared to CT, in line with previous studies where LY supplementation alleviated the severity of diarrhea in piglets [26,27].

Although dietary LY administration during mid-late gestation and lactation of primiparous sows had little effect on sows and piglets performance, significant and interesting results were observed on the immune status of the dams and their offspring. Dietary supplementation with LY increased serum concentrations of IgA and IgG in sows at farrowing and weaning, and in piglets in the post-weaning stage on days 14 and 42. Similarly, a significant increase in plasma concentration of IgG at day 1 of lactation was observed in sows with LY supplementation during late gestation [12]. Jang et al. (2013) [8] reported that LY supplementation to sows’ diets was associated with an increase of plasma IgG concentration of their progeny, whereas no effects were observed in plasma IgA concentrations of sows and piglets, which suggested that LY stimulated the immune system of sows, and subsequently more IgG could be contained in colostrum and be transferred to piglets. Yeast and yeast products derived from *S. cerevisiae* are immunomodulating compounds that can have positive effects both directly and indirectly on the immune system [28,29], because LY contains β-glucan and oligosaccharides that exert immune-modulating effects [29,30]. In line with these results, supplementation of sow diets with MOS derived from yeast cell wall increased the concentration of IgM [31]. Based on the results of our study, the increasing IgA and IgG levels in piglets may suggest that LY administration during middle–late gestation and the lactation period can improve the passive immunity of piglets, thus favoring their health.

Farrowing and weaning usually bring severe oxidative stress to sows and piglets, and yeast and yeast-derived product have recognized antioxidant properties [32]. MDA and SOD are classic biomarkers to reflect the intensity of lipid peroxidation. In this study, no significant differences were found in serum concentrations of MDA, T-AOC, and SOD activity, both in primiparous sows and weaned piglets. However, serum activity of CAT in sows at weaning was improved by LY supplementation, and a mild increase in SOD was observed in piglets at day 14 post-weaning. In line with these results, dietary LY supplementation increased serum SOD activity and decreased serum MDA concentration in post-weaning piglets at days 7 and 21, while at the same time it increased serum CAT level at day 21 [33]. The mechanism of LY on alleviating oxidative stress might be related to the beneficial effect of β-glucan contained in LY, which can enhance antioxidant capacity and reduce the oxidative damage of lymphocytes through different enzymatic and non-enzymatic systems.

## 5. Conclusions

In conclusion, live yeast addition in the diets of primiparous sows during middle–late gestation and the whole lactation period significantly increased the immunity of sows and their offspring, therefore, favoring maternal and progeny health.

## Figures and Tables

**Table 1 animals-12-01315-t001:** Ingredient compositions and nutrient levels of gestation and lactation diets (%, as-fed basis).

Items	Gestation	Lactation
Ingredients		
Corn	67.69	62.33
Soybean meal, 46% CP	14.00	19.00
Fish meal	-	3.00
Extruded soybean	-	6.70
Wheat bran	14.00	4.57
Soybean oil	1.60	1.00
Sodium chloride	0.40	0.30
Limestone	1.13	1.15
Calcium dihydrogen phosphate	0.57	0.95
Phytase	0.04	-
Choline chloride, 60%	0.07	0.10
L-Lysine HCl, 98.5%	-	0.35
L-Threonine	-	0.05
Vitamin and mineral premix ^1^	0.50	0.50
Total	100.00	100.00
Analyzed nutrient levels		
Crude protein	14.6	17.4
Total Calcium	0.62	0.75
Total Phosphotus	0.50	0.64
Calculated nutrient levels		
Metabolizable energy, kcal/kg	3108	3183
Lysine, %	0.63	0.95
Methionine + cysteine, %	0.48	0.56
Threonine, %	0.51	0.65
Tryptophan, %	0.16	0.20

^1^ The premix supplied the following vitamins and trace minerals per kilogram of diet: Cu, 15 mg; I, 0.3 mg; Mn, 50 mg; Se, 0.3 mg; Fe, 80 mg; Zn, 100 mg; 25,000 IU vitamin A; 5000 IU vitamin D_3_; 50 IU vitamin E; 2.5 mg vitamin K; 0.2 mg biotin; 1.0 mg vitamin B_1_; 8.0 mg vitamin B_2_; 3.0 mg vitamin B_6_; 0.020 mg vitamin B_12_; 15.0 mg niacin; 12.5 mg pantothenic acid; 1.50 mg folacin.

**Table 2 animals-12-01315-t002:** Ingredient compositions and nutrient levels of nursery diets in phases 1 and 2 (%, as-fed basis).

Items	Phase 1 d 0–14	Phase 2 d 14–42
Ingredients		
Corn	16.45	21.17
Extruded corn	32.00	40.00
Soybean meal, 46% CP	14.00	17.50
Extruded soybean	11.50	6.00
Fish meal	5.60	3.00
Dried whey	15.00	5.00
Soybean oil	1.00	1.20
Bran	-	1.50
Calcium dihydrogen phosphate	0.40	0.60
Limestone	0.75	0.90
Sodium chloride	0.30	0.30
Choline chloride, 60%	0.05	0.05
L-Lysine HCl, 98.5%	1.20	1.08
DL-Methionine	0.09	0.08
L-Threonine	0.27	0.24
Tryptophan	0.02	0.01
Phytase	0.02	0.02
Acidifier	0.35	0.35
Vitamin and mineral premix ^1^	1.00	1.00
Total	100.00	100.00
Analyzed nutrient levels		
Crude protein	19.62	17.62
Total Calcium	0.79	0.74
Total Phosphotus	0.66	0.62
Calculated nutrient levels		
Metabolizable energy, kcal/kg	3400	3350
Lysine, %	1.30	1.15
Methionine, %	0.38	0.34
Threonine, %	0.76	0.68
Tryptophan, %	0.21	0.19

^1^ The premix provided the following per kg of diets: niacin, 38.4 mg; calcium pantothenate, 25 mg; folic acid, 1.68 mg; biotin, 0.16 mg; vitamin A, 35.2 mg; vitamin B1, 4 mg; vitamin B_2_, 12 mg; vitamin B_6_, 8.32 mg; vitamin B_12_, 4.8 mg; vitamin D_3_, 7.68 mg; vitamin E, 128 mg; vitamin K_3_, 8.16 mg; Cu, 125 mg; Zn, 110 mg; Se, 0.19 mg; Fe, 171 mg; Co, 0.19 mg; Mn, 42.31 mg; I, 0.54 mg.

**Table 3 animals-12-01315-t003:** The effects of dietary live yeast supplementation during middle–late gestation and lactation on body weight, backfat thickness, and feed intake of primiparous sows.

Items	CT	LYT	*p*-Value
Number of sows			
Day 60 of gestation	8	8	-
During lactation	6	7	-
Body weight, kg			
Day 60 of gestation	188.0 ± 5.8	188.6 ± 7.1	NS
Day 110 of gestation	234.3 ± 5.5	233.3 ± 4.8	NS
At weaning	185.6 ± 5.2	192.5 ± 4.6	NS
Average backfat thickness, mm			
Day 60 of gestation	17.80 ± 0.20	17.80 ± 0.25	NS
Day 110 of gestation	18.13 ± 0.31	18.47 ± 0.27	NS
At weaning	16.84 ± 0.23	16.33 ± 0.20	NS
Average feed intake/sows, kg			
Day 60 of gestation to farrowing	151.5 ± 6.6	159.03 ± 5.8	NS
During lactation	111.2 ± 14.5	112.5 ± 12.7	NS
Average daily feed intake/sows, kg			
Day 60 of gestation to farrowing	2.76 ± 0.11	2.82 ± 0.09	NS
DDuring lactation	3.93 ± 0.40	4.15 ± 0.35	NS

CT = control group, basal diet; LYT = live yeast treatment, basal diet + 425 mg/kg live yeast; NS = no significance. Values are expressed as Mean ± Standard Error.

**Table 4 animals-12-01315-t004:** The effects of dietary live yeast supplementation during middle–late gestation and lactation on reproduction performance of primiparous sows.

Items	CT	LYT	*p*-Value
Number of sows	6	7	
Gestation lenght	114.8 ± 0.65	116.3 ± 0.57	NS
At farrowing			
Total born	14.50 ± 1.21	12.71 ± 1.06	NS
Born alive	13.00 ± 0.94	10.57 ± 0.82	0.10
Still born	0.83 ± 0.59	1.14 ± 0.51	NS
Mummy	0.67 ± 0.46	1.00 ± 0.40	NS
Average birth weight, kg	1.42 ± 0.14	1.67 ± 0.12	NS
Average litter weight, kg	17.81 ± 1.49	17.33 ± 1.30	NS
Percentage of average birth weight < 1.0 kg	10.55 ± 5.68	2.22 ± 4.97	NS
Cross-fostering			
Average litter size	12.00 ± 0.83	10.86 ± 0.73	NS
Average body weight, kg	1.43 ± 0.14	1.66 ± 0.12	NS
Average litter weight, kg	17.08 ± 1.35	17.77 ± 1.18	NS
During lactation			
Average lactation days	28.17 ± 1.18	27.14 ± 1.04	NS
Average body weight gain of piglets, kg	5.82 ± 0.90	6.30 ± 0.79	NS
Average daily gain of piglets, g	204.5 ± 29.0	228.8 ± 25.4	NS
Mortality of piglets, %	2.56 ± 2.17	4.72 ± 1.90	NS
At weaning			
Average litter size	11.67 ± 0.58	10.29 ± 0.51	NS
Average litter weight, kg	84.27 ± 6.73	80.30 ± 5.89	NS
Average body weight, kg	7.25 ± 0.97	7.96 ± 0.85	NS

CT = control group, basal diet; LYT = live yeast treatment, basal diet + 425 mg/kg live yeast; NS = no significance. Values are expressed as Mean ± Standard Error.

**Table 5 animals-12-01315-t005:** Growth performance and diarrhea incidence of weaned piglets born to CT and LYT primiparous sows.

Items	CT	LYT	*p*-Value
BW, kg			
Day 0	7.36 ± 0.58	8.01 ± 0.69	0.02
Day 14	8.68 ± 0.68	9.18 ± 0.91	NS
Day 28	11.84 ± 0.61	12.23 ± 1.14	NS
Day 42	17.34 ± 0.79	19.24 ± 1.87	NS
ADG, g			
Day 0–14	95 ± 17	84 ± 21	NS
Day 14–28	225 ± 27	218 ± 22	NS
Day 28–42	393 ± 26	501 ± 56	NS
Day 0–42	238 ± 14	267 ± 29	NS
ADFI, g			
Day 0–14	212 ± 9	248 ± 29	NS
Day 14–28	668 ± 46	666 ± 51	NS
Day 28–42	719 ± 62	950 ± 113	0.10
Day 0–42	533 ± 26	621 ± 49	NS
Gain: Feed			
Day 0–14	0.45 ± 0.08	0.32 ± 0.06	NS
Day 14–28	0.34 ± 0.03	0.33 ± 0.02	NS
Day 28–42	0.55 ± 0.02	0.56 ± 0.09	NS
Day 0–42	0.44 ± 0.01	0.43 ± 0.04	NS
Diarrhea incidence, %			
Day 0–14	16.93	13.03	0.10

BW = body weight; ADG = average daily gain; ADFI = average daily feed intake; CT = control group, basal diet; LYT = live yeast treatment, basal diet + 425 mg/kg live yeast; NS = no significance. Values are expressed as Mean ± Standard Error.

**Table 6 animals-12-01315-t006:** The effect of live yeast supplementation during middle–late gestation and lactation on the compositions of colostrum and milk (%).

Items	CT	LYT	*p*-Value
Colostrum			
Fat	4.58 ± 0.49	5.06 ± 0.43	NS
Protein	14.76 ± 1.36	14.83 ± 1.19	NS
Lactose	2.59 ± 0.29	2.52 ± 0.25	NS
Total solids	21.37 ± 0.88	20.41 ± 0.77	NS
Solids-not-fat	16.79 ± 0.60	15.35 ± 0.52	NS
Milk			
Fat	7.74 ± 0.47	7.47 ± 0.41	NS
Protein	5.09 ± 0.13	4.94 ± 0.12	NS
Lactose	5.74 ± 0.09	5.97 ± 0.08	0.10
Total solids	17.77 ± 1.05	16.39 ± 0.92	NS
Solids-not-fat	10.03 ± 0.86	8.91 ± 0.75	NS

CT = control group, basal diet; LYT = live yeast treatment, basal diet + 425 mg/kg live yeast; NS = no significance. Values are expressed as Mean ± Standard Error.

**Table 7 animals-12-01315-t007:** The effects of live yeast supplementation during middle–late pregnancy and lactation period on serum immunological parameters (mg/mL) in primiparous sows and weaned piglets.

Items	CT	LYT	*p*-Value
Primiparous sows			
At farrowing			
IgA	2.88 ± 0.24	4.91 ± 0.21	<0.01
IgG	153.95 ± 3.67	168.46 ± 3.28	0.03
IgM	12.20 ± 3.95	22.20 ± 3.56	NS
At weaning			
IgA	2.79 ± 0.48	5.27 ± 0.36	0.02
IgG	126.07 ± 8.50	156.69 ± 6.40	0.05
IgM	19.45 ± 4.19	25.48 ± 3.15	NS
Weaned piglets			
Day 0			
IgA	1.01 ± 0.11	1.54 ± 0.09	0.02
IgG	56.55 ± 8.39	75.01 ± 6.32	NS
IgM	4.89 ± 0.38	7.31 ± 0.28	0.01
Day 14			
IgA	2.03 ± 0.17	2.93 ± 0.10	<0.01
IgG	59.30 ± 5.66	82.38 ± 11.59	0.02
IgM	10.77 ± 1.04	16.20 ± 2.22	0.06
Day 42			
IgA	2.74 ± 0.10	3.54 ± 0.14	<0.01
IgG	85.42 ± 4.56	105.63 ± 5.77	0.01
IgM	28.30 ± 4.69	29.96 ± 8.09	NS

CT = control group, basal diet; LYT = live yeast treatment, basal diet + 425 mg/kg live yeast; NS = no significance. Values are expressed as Mean ± Standard Error.

**Table 8 animals-12-01315-t008:** The effects of live yeast supplementation during late pregnancy and lactation period on antioxidant parameters in the plasma of primiparous sows and weaned piglets.

Items	CT	LYT	*p*-Value
Primiparous sows			
At farrowing			
MDA, nmol/mL	2.91 ± 1.02	1.89 ± 0.89	NS
SOD, U/mL	70.95 ± 4.75	73.94 ± 4.26	NS
T-AOC, nmol/mL	1.48 ± 0.03	1.50 ± 0.02	NS
CAT, U/mL	2.90 ± 0.24	3.27 ± 0.21	NS
At weaning			
MDA, nmol/mL	2.67 ± 0.64	1.99 ± 0.48	NS
SOD, U/mL	23.29 ± 1.16	25.64 ± 0.87	NS
T-AOC, nmol/mL	1.67 ± 0.04	1.70 ± 0.03	NS
CAT, U/mL	4.81 ± 0.06	5.53 ± 0.05	<0.01
Weaning piglets			
Day 0			
MDA, nmol/mL	2.55 ± 0.36	2.44 ± 0.27	NS
SOD, U/mL	41.70 ± 2.71	39.52 ± 2.04	NS
T-AOC, nmol/mL	1.16 ± 0.08	1.23 ± 0.06	NS
CAT, U/mL	5.51 ± 0.97	5.71 ± 0.73	NS
Day 14			
MDA, nmol/mL	1.86 ± 0.24	1.91 ± 0.12	NS
SOD, U/mL	28.28 ± 1.07	31.10 ± 1.39	0.10
T-AOC, nmol/mL	1.27 ± 0.04	1.29 ± 0.06	NS
CAT, U/mL	4.80 ± 0.86	4.09 ± 0.66	NS
Day 42			
MDA, nmol/mL	1.40 ± 0.16	1.55 ± 0.19	NS
SOD, U/mL	28.69 ± 1.79	28.81 ± 0.77	NS
T-AOC, nmol/mL	1.33 ± 0.05	1.37 ± 0.04	NS
CAT, U/mL	2.91 ± 0.22	4.19 ± 0.74	NS

MDA = malondialdehyde; SOD = superoxide dismutase; T-AOC = total antioxidant capacity; CAT = catalase; CT = control group, basal diet; LYT = live yeast treatment, basal diet + 425 mg/kg live yeast; NS = no significance. Values are expressed as Mean ± Standard Error.

## Data Availability

The data presented in this study are available on request from the corresponding author.

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
