# Peer review of "Live Yeast Supplementation in Gestating and Lactating Primiparous Sows Improves Immune Response in Dams and Their Progeny"

_animals, 2022, doi:10.3390/ani12101315_

Round 1

Reviewer 1 Report

Tian Xia et al. reported live yeast supplementation in  primiparous sows during gestating and lactating  improves immune response in dams and their progeny. Presented manuscript is interesting, but I have seen a few points that need to be clarified.

  1. Primiparous sows were the subject of the presented research.In which oestrus cycle (after puberty) were sows mated or inseminated?At what body weight have the gilts reached sexual maturity?
  2. One of the parameters tested in the manuscript under review was immunological parameters. Were the eligible animals healthy? It is known that, for example, Porcine parvovirus (PPV), Porcine reproductive and respiratory syndrome virus (PRRS), Classical swine fever virus (CSF) and Swine influenza virus (SI) can affect both reproductive characteristics and parameters of the immune system. Were sows tested for the presence / exclusion of these diseases? Was the herd from which these animals were derived free from these diseases?
  3. In Tables 3-8, the authors reported the SEM value. Which CT or LYT group does this SEM refer to? The SEM value for both the CT and LYT groups should be reported. The easiest way to do this is to use the ±- symbol, e.g. 122 ±15. No additional columns in the tables.
  4. Linie176-177 The authors assumed: "Differences were declared significant at p ≤ 0.05, while 0.05 <p ≤ 0.10 was considered to indicate a trend in the data." I don't understand why in Tables 3-8 there are presented pValue for non-statistically significant data. What was the purpose of presentation these values?
  5. In Table 4, the parameters of the litters after birth (total born, born alive, still born and mummy) for the LYT group are worse / weaker than in the CT group. The authors, describing the results included in Table 4 (lines 182-185), reported that the parameters in both groups were similar. They did not mention, however, that the "Born alive" parameter was showing a trend whose assumptions were described on lines 176-177. I suppose that if the study groups were much more numerous, these differences would be statistically significant.
  6. Why do the authors provide p> 0.05 for the described statistically insignificant data (it concerns the entire manuscript)? This confuses the reader.
  7. The authors investigated the occurrence of diarrhea in piglets. Has diarrhea also occurred in the sows? Maybe the higher immunoglobulin values in piglets and sows from the LYT group are not the health-promoting effect of live yeast feeding, but the digestive tract disorders?
  8. Moreover, the number of animals analyzed is too small.

Author Response

Tian Xia et al. reported live yeast supplementation in primiparous sows during gestating and lactating improves immune response in dams and their progeny. Presented manuscript is interesting, but I have seen a few points that need to be clarified.

Response (R): We would highly appreciate the comments provided by the reviewer to improve the quality of our manuscript. We have revised the manuscript according to the comments pointed out by the reviewer.

1. Primiparous sows were the subject of the presented research. In which oestrus cycle (after puberty) were sows mated or inseminated? At what body weight have the gilts reached sexual maturity?

R: All the gilts were inseminated in the second oestrus cycle and reached sexual maturity at 140~150 kg. Please check Line 75-77.

2. One of the parameters tested in the manuscript under review was immunological parameters. Were the eligible animals healthy? It is known that, for example, Porcine parvovirus (PPV), Porcine reproductive and respiratory syndrome virus (PRRS), Classical swine fever virus (CSF) and Swine influenza virus (SI) can affect both reproductive characteristics and parameters of the immune system. Were sows tested for the presence / exclusion of these diseases? Was the herd from which these animals were derived free from these diseases?

R: Thank you very much for your comments. We did not detect the diseases for the gilts. All the gilts used in this study were from a health breeding farm (PRRS and CSF negative). These gilts did not show any sign of infection for the viruses as mentioned. Moreover, the gilts were randomly assigned into two treatments to minimize the effects of other factors including virus infection.

3. In Tables 3-8, the authors reported the SEM value. Which CT or LYT group does this SEM refer to? The SEM value for both the CT and LYT groups should be reported. The easiest way to do this is to use the ±- symbol, e.g. 122 ±15. No additional columns in the tables.

R: We inserted the SE value of each treatment to replace SEM according to your comments, please check the Tables.

4. Linie176-177 The authors assumed: "Differences were declared significant at p ≤ 0.05, while 0.05 <p ≤ 0.10 was considered to indicate a trend in the data." I don't understand why in Tables 3-8 there are presented pValue for non-statistically significant data. What was the purpose of presentation these values?

R: Thank you very much for your suggestion. We use “NS” to replace the P values that are higher than 0.10 in the Tables, please check them.

5. In Table 4, the parameters of the litters after birth (total born, born alive, still born and mummy) for the LYT group are worse / weaker than in the CT group. The authors, describing the results included in Table 4 (lines 182-185), reported that the parameters in both groups were similar. They did not mention, however, that the "Born alive" parameter was showing a trend whose assumptions were described on lines 176-177. I suppose that if the study groups were much more numerous, these differences would be statistically significant.

R: We agree on reviewer’s suppose that the difference would be significant if the replicates were more than we used in this trial. Actually, we slightly described “born alive was numerically lower but showed no significant difference”, please check Line 204-206. However, it was reported that the litter size in sows was mainly determined by the fertilization rate and prenatal death occurring in the early pregnancy [15] (Line 291-293), additionally, results can be affected by many factors, such as management, amount of yeast, number of animals, and so on. Thus, on the other hand, if the study groups were much more numerous, the differences between two groups may be canceled out. Therefore, more researches should be conducted.

6. Why do the authors provide p> 0.05 for the described statistically insignificant data (it concerns the entire manuscript)? This confuses the reader.

R: Thank you very much for your comments. In our study, we observed the main significant results concerning plasma immunoglobulin and indeed we mentioned them in the abstract. In addition, we also observed some numerical improvements of LY supplementation on lactation performance of sows and diarrhea incidence of piglets under the background with limited sows and piglets, which might provide ideas and hypothesis for the further studies.

7. The authors investigated the occurrence of diarrhea in piglets. Has diarrhea also occurred in the sows? Maybe the higher immunoglobulin values in piglets and sows from the LYT group are not the health-promoting effect of live yeast feeding, but the digestive tract disorders?

R: Thank you very much for your comments. The gilts in our study were healthy and without any diarrhea phenomenon during the whole trial.

8. Moreover, the number of animals analyzed is too small.

R: We completely agree on your concern. Limited animals were used in our study due to many reasons such as the budget, and we would increase the number of gilts in the further study based on the observed tendencies in our current study.

Reviewer 2 Report

Comments on the manuscript:

“Live yeast supplementation in gestating and lactating primiparous sows improves immune response in dams and their progeny”

The addition of live yeast to the gestation and lactation diets of sows is known to have positive effects. However, the residual effects of live yeast supplementation during gestation and lactation on primiparous sows and their offspring have yet to be elucidated. The manuscript presented here concerns a study of the effects of live yeast supplementation in intermediate and late gestation and lactation diets on reproduction, lactation and the immune system of primiparous sows. The study also concerns the effects on the growth and immunity of the offspring. In conclusion, the study shows that the addition of live yeasts in the feed of primiparous sows during gestation and lactation has the effect of increasing the immunity of sows and their offspring.

This very clear and well-conducted work brings interesting elements to the knowledge of the effects of live yeast supplementation in primiparous sows, which could have repercussions at the breeding level. I think this text can be published with just a few minor corrections.

Page 5, lines 158-160 “Concentrations of immunoglobulin A (IgA), … were detected by corresponding commercial kits”: it would be useful to explain the method used concisely: the reference to the kit's instructions seems to me insufficient for a scientific article.

Page 5, lines 162-164: “…were measured by a spectrophotometer (Lengguang SFZ1606017568, Shanghai, China), according to the instructions of the corresponding reagent kits”: it would be useful to explain the method used concisely: see above.

Author Response

The addition of live yeast to the gestation and lactation diets of sows is known to have positive effects. However, the residual effects of live yeast supplementation during gestation and lactation on primiparous sows and their offspring have yet to be elucidated. The manuscript presented here concerns a study of the effects of live yeast supplementation in intermediate and late gestation and lactation diets on reproduction, lactation and the immune system of primiparous sows. The study also concerns the effects on the growth and immunity of the offspring. In conclusion, the study shows that the addition of live yeasts in the feed of primiparous sows during gestation and lactation has the effect of increasing the immunity of sows and their offspring.

This very clear and well-conducted work brings interesting elements to the knowledge of the effects of live yeast supplementation in primiparous sows, which could have repercussions at the breeding level. I think this text can be published with just a few minor corrections.

R: Thank you very much for the comments provided by the reviewer to improve the quality of our manuscript. We modified the manuscript according to reviewer’s suggestion.

Page 5, lines 158-160 “Concentrations of immunoglobulin A (IgA), … were detected by corresponding commercial kits”: it would be useful to explain the method used concisely: the reference to the kit's instructions seems to me insufficient for a scientific article.

R: We added the information to explain the method. Please check Line 160-171.

Page 5, lines 162-164: “…were measured by a spectrophotometer (Lengguang SFZ1606017568, Shanghai, China), according to the instructions of the corresponding reagent kits”: it would be useful to explain the method used concisely: see above.

R: Thank you very much for your suggestion, and we added the information. Please check Line 176-183.

Round 2

Reviewer 1 Report

The authors made some corrections and most of the answers are satisfactory. However, there are still some issues in the manuscript that require clarification.

The authors reported the body weight of all sows 188.3±17.7 kg and backfat thickness 17.8±0.62 mm (line 78), however, this does not agree with the data in Table 3. As for the authors, the values SD 5.8 and 7.1 kg (Table 3) ) came out 17.7 kg (line 78). It is similar with backfat thickness: in table 3 SD are 0.20 and 0.25, in the text is 0.62 mm (line 78)? The data in Table 3, after dividing the sows into groups, indicate that the CT and LYT groups were much more even than the authors described in the manuscript (Materials and Methods).

Lines 222-233: The authors describe the results contained in both tables 4 and 5. Unfortunately, they do not indicate exactly which data is in which table, which makes it difficult to read this fragment of the text. This difficulty is exacerbated by the appearance of one reference to the table, which is unfortunately incorrect. The authors cite table 5, and they should 4. Also, the authors provide percentages (lines 230-232) that are not in any table, even though they performed statistical analyzes for this data.

The authors in the tables improved the given p-Value, while the test still has statistically insignificant p values, e.g. line 225.

Author Response

The authors made some corrections and most of the answers are satisfactory. However, there are still some issues in the manuscript that require clarification.

Response (R): Thanks a lot for the comments provided by the reviewer to avoid the confusion in our manuscript. We modified the parts according to the comments as below.

1. The authors reported the body weight of all sows 188.3±17.7 kg and backfat thickness 17.8±0.62 mm (line 78), however, this does not agree with the data in Table 3. As for the authors, the values SD 5.8 and 7.1 kg (Table 3) ) came out 17.7 kg (line 78). It is similar with backfat thickness: in table 3 SD are 0.20 and 0.25, in the text is 0.62 mm (line 78)? The data in Table 3, after dividing the sows into groups, indicate that the CT and LYT groups were much more even than the authors described in the manuscript (Materials and Methods).

R: Thank you very much for pointing out this issue. Actually, the values in the Line 78 were SD of body weight (17.7) and backfat thickness (0.62), and the values in the Table 3 were SE. Thus, we changed the relevant data using SE as 6.4 and 0.22 for body weight and backfat thickness, respectively. Please check Line 78.

2. Lines 222-233: The authors describe the results contained in both tables 4 and 5. Unfortunately, they do not indicate exactly which data is in which table, which makes it difficult to read this fragment of the text. This difficulty is exacerbated by the appearance of one reference to the table, which is unfortunately incorrect. The authors cite table 5, and they should 4. Also, the authors provide percentages (lines 230-232) that are not in any table, even though they performed statistical analyzes for this data.

R: Thank you very much for the correction. We modified the sentences to clarify the results of body weight at weaning in tables 4 and 5, please check Line 226. In addition, we modified the sentences concerning the percentages to indicate the differences of ADFI from day 28 to 42 and diarrhea incidence from day 0 to 14 between CT and LYT groups, please check Line 231 to 234.

3. The authors in the tables improved the given p-Value, while the test still has statistically insignificant p values, e.g. line 225.

R: We modified the relevant interpretation and checked the whole section to avoid this mistake. Please check Line 226.